# Intuitive or deliberative dishonesty: The effect of abstract versus concrete victim

**Jiayu Cheng**[1], **Haoran Wang**[1], **Yue Liu**[1], **Chongxiang Wang**[1], **Qingzhou Sun**[2]*, **Bruno Verschuere**[3], **Liyang Sai**[1,4]*

**1** Department of Psychology, Hangzhou Normal University, Hangzhou, China, **2** School of Management, Zhejiang University of Technology, Hangzhou, China, **3** Department of Clinical Psychology, University of Amsterdam, Amsterdam, the Netherlands, **4** Zhejiang Philosophy and Social Science, Laboratory for Research in Early Development and Childcare, Hangzhou Normal University, Hangzhou, China

* sunqingzhou2008@163.com (QS); liyangsai@hznu.edu.cn (LS)

## Abstract

There has been ongoing debate over whether people are intuitively honest or intuitively dishonest. A recent social harm account was proposed to address this debate: dishonesty is intuitive when cheating inflicts harm on an abstract other while honesty is intuitive when cheating inflicts harm on a concrete other. This pre-registered and well-powered study ($n = 764$) aims to directly test this account by using a time pressure manipulation. Specifically, we examined whether time pressure (versus self-paced conditions) would lead to increased cheating depending on whether the harmed party was concrete or abstract. The results showed no significant effect of time pressure on cheating behavior. However, the harm-type manipulation produced findings that contradicted those reported in previous studies. Given the low replication rates and reliance on controversial experimental manipulations in this area, our findings underscore the importance of further pre-registered research to rigorously evaluate the roles of time pressure and social harm in shaping intuitive (dis)honesty.

## 1. Introduction

Although honesty is valued across countries, dishonesty is ubiquitous in everyday life, including financial fraud in large corporations, tax evasion of institutions or celebrities, and academic cheating by individuals. And some researchers are even being accused of cheating in their research about dishonesty [1]. These behaviors carry significant economic costs and erode trust in relationships and institutions [2]. Therefore, researchers from different disciplines paid much attention to how and why individuals cheat. One of the interesting questions researchers are paying attention to is whether honesty or dishonesty is intuitive when dishonesty can obtain profits. Yet, there are opposing theoretical perspectives and conflicting empirical data.

*Grace theory* proposes that honesty is an intuitive behavior, with individuals needing cognitive control to inhibit their honest tendency when making dishonest

**Data availability statement:** All data and materials associated with this study are publicly available at Open Science Framework (https://osf.io/vgmsp/overview).

**Funding:** This work was supported by the National Natural Science Foundation of China (32271111, U1736125 to L. Sai); the Science and Technology Innovation 2030-"Brain Science and Brain-like Research" Major Project (Grant/Award Number: 2022ZD0210800). The funders had no role in study design, data collection and analysis, decision to publish, or preparation of the manuscript. There was no additional external funding received for this study.

**Competing interests:** The authors have declared that no competing interests exist.

decisions. This theory is supported by a number of studies. For example, by making participants' decision-making process under time pressure [3,4] or increasing participants' cognitive load by using a distracting task [5], researchers found that people are more likely to be honest when their cognitive resources are limited [5,6]. These findings demonstrate that dishonesty is cognitively demanding, while honesty is a more intuitive response. Contrarily, *Will theory* propose that dishonesty is intuitive, and individuals need cognitive control to resist the temptation to cheat [7,8]. For example, Shalvi et al. [9] found that time pressure increased cheating (but see [10]). Other experimental works have similarly revealed that through cognitive load [11–13], mental or physical depletion [14–16], priming of intuition concepts [17] may increase self-serving dishonesty. These findings suggest that dishonesty comes naturally, whereas honesty requires overcoming the initial tendency to cheat.

The *social harm theory* [18] can integratively explain the seemingly opposing research findings. It proposes that dishonesty is intuitive when it harms an abstract, vaguer entity (e.g., the experimental budget), while dishonesty is deliberative when it harms a concrete other (e.g., a participant). The study by Pitesa et al. [19] provided direct evidence for this theory. In their study, a subset of participants initially engaged in a cognitive control task, followed by participation in a cheating task where they had the opportunity to cheat for a monetary reward. Subsequently, participants were randomly assigned to either a concrete harm group or an abstract harm group. The results showed that in these cognitively depleted participants, cheating decreased when it caused harm to another participant, and increased when it did not harm a concrete other person (their behaviour would not affect the other participant's reward). However, no such difference was observed in the group without cognitive impairment. Consistent with findings from Pitesa et al. [19], the meta-analysis by Köbis et al. [18] found that when dishonesty harms abstract others, promoting intuition causes more people to cheat. However, when dishonesty inflicts harm on concrete others, promoting intuition has no significant effect on dishonesty.

Thus, the social consequences of dishonesty could be a promising key to the riddle of intuition's role in honesty. However, the empirical evidence supporting this theory is limited. First, Pitesa et al. [19] provided evidence about this theory, but their study was not pre-registered, and their main findings were not well powered (given the observed effect size ($\eta_p^2$) being 0.073 to 0.094). Thus, it is important to have well-powered and pre-registered studies to test it again [20]. Second, while meta-analyses serve as a valuable source of research synthesis, an inherent limitation is that they rely on unbiased input. There are indications of publication bias in this literature [21]. Moreover, many studies in the Köbis et al. meta-analysis rely on manipulations (e.g., ego-depletion, behavioral priming) that are at the heart of the replication crisis [22,23].

Given the above limited findings, the present pre-registered study aims to directly test the *social-harm theory* with time-pressure for intuitive dishonesty. We examined whether time pressure (as compared to a self-paced condition) leads to more cheating when there is an abstract victim or there is a concrete victim. We employed an online dice-rolling task modified from Shalvi et al. [9] (Experiment 2). Participants

rolled a die three times, reporting the first outcome to determine their compensation—the higher the roll, the greater the payment. We manipulated harm type, following the approach of Pitesa et al. [19]. In the concrete victim condition, participants were told that another participant would be paid from the same pot (a fixed amount), which meant their earnings would decrease the payment of another participant's share of that fixed amount. In the abstract victim condition, no other participants would be paid from the same pot as the participant (These two conditions align with the interpersonal impact salient and not salient conditions labels, respectively, as defined in Pitesa et al. [19], Experiment 2, but we adopt the terminology that is consistent with social harm theory proposed by Köbis et al. [18]). To manipulate time pressure, we followed the procedure used in Experiment 2 of Shalvi et al. [9]. Participants in the time pressure condition had 13 seconds to complete the task and report their initial dice roll, including the 8-second reporting window used by Shalvi et al. [9], with 5 seconds needed for dice rolling in our online setting. In the self-paced condition, participants had unlimited time to complete the task and submit their response.

As exploratory analyses, we also considered the influence of individual traits on cheating behavior. For instance, Xu and Ma observed that individuals with a high moral identity typically exhibit an intuitive honesty, while those with a low moral identity tend towards dishonesty [24]. Furthermore, Bacon et al. [25] reported that reward sensitivity traits are inversely correlated with academic dishonesty in intrinsically motivated students, yet positively correlated in those motivated by grade achievement. Additionally, empirical evidence suggests that high Machiavellian individuals are more prone to lying and experience less guilt post-deceit [26]. Accordingly, the second aim of the present study is to explore whether traits such as Moral Identity, Reward Sensitivity, and Machiavellianism influence an individual's intuitive inclination towards honesty or dishonesty.

Based on *the social-harm account of intuitive dishonesty* [9,27], we expected an interaction between time pressure and harm type. Specifically, we expected that time pressure makes participants more likely to cheat in the harm abstract victim condition, while time pressure makes participants more likely to be honest in the harm concrete victim condition. Additionally, we explore how individual traits such as Moral Identity, Reward Sensitivity, and Machiavellianism influence participants' intuitive tendencies toward honesty or dishonesty.

## 2. Methods

The study was preregistered (https://aspredicted.org/M9Q_8NF). This study was approved by the Ethics Committee of Hangzhou Normal University (NO. 20220301). All methods were performed in accordance with the relevant guidelines and regulations. All participants read and agreed to the informed consent form before beginning the online task and were informed that they could withdraw at any time. The research was conducted in accordance with the Declaration of Helsinki. Data was collected between July 2022 and October 2022.

### 2.1. Participants

G*Power 3.1 software was used for prior power analysis to determine the sample size in this study [28] with Power $(1-\beta)$ set at 0.8 and $\alpha = 0.05$, which showed that to detect a significant interaction effect in a binary logistic regression with a medium effect size (OR = 1.506, computed by using our pilot data), 763 participants would be required (as preregistered). Specifically, participants were recruited from Hangzhou Normal University and Zhejiang University through both electronic and printed advertisements, and from Shenzhen University and Southwest University via online subject pools. Recruitment information was also disseminated via WeChat Moments. Interested individuals could access the study by scanning a QR code provided in the digital or printed posters, which directed them to the online game platform. A total of 1151 participants were recruited in China online, and exclusions were monitored until reaching a minimum of 763 valid participants. Following pre-registration, we excluded 186 participants (16.15%) with missing data in the die-rolling task, 11 participants (0.96%) who reported smaller outcomes than reality, and 190 participants (16.51%) didn't complete the task and automatically excluded. Finally, 764 (66.38%) valid data was obtained, with 191 participants in each group (time pressure

with abstract victim, $M=23.45$, $SD=4.70$, 58.64% female; time pressure with concrete victim, $M=23.47$, $SD=4.56$, 50.79% female; self-paced with abstract victim, $M=23.19$, $SD=4.48$, 53.93% female; and self-paced with concrete victim, $M=23.40$, $SD=4.57$, 55.50% female), no significant differences in age or gender among the four groups.

## 2.2. Procedures

*Die-rolling Task*. A die-rolling task was used to measure participants' cheating behavior. This task was adapted from the study by Shalvi et al. [9], with the only difference being that it was implemented online. Participants were instructed to roll three dice sequentially by pressing a button and then report the outcome of the first roll by pressing a number between 1 and 6. They were told that their payment would depend on the reported number—the higher the number, the greater the reward.

The experiments were conducted on participants' smartphones, using the *WeChat Mini Program* for stimulus presentation. After entering the program, participants were first asked to fill in their personal information such as gender and age. Then, participants were told that there was another participant who would take part in the experiment with them simultaneously. However, unknown to the participants, the other participant was not real.

Participants were randomly assigned into one of the four conditions (time pressure with concrete victim, time pressure with abstract victim, self-paced with concrete victim, and self-paced with abstract victim). In each condition, participants were told to play a die-rolling task. Specifically, participants were asked to roll three dice one by one, and only report the outcome of the first die-rolling (To ensure that all participants have the opportunity to lie, there will be no point 6 on the first die roll). Their payoff was dependent on what they reported in the game, with higher reported numbers resulting in greater payments (e.g., reporting "1" led to ¥1 yuan, while reporting "6" led to ¥6 yuan). In the time pressure condition, participants only had 13 seconds in total to complete the 3 times dice roll and report their outcomes, whereas the self-paced condition had no time limit. In the concrete victim condition, participants were informed of sharing a ¥7 yuan bonus with other participants, which means the more payoff you receive, the less payoff the other participants would obtain. Conversely, in the abstract victim condition, participants were informed that they were playing the game independently and would not be sharing the bonus with anyone else. Rule-check questions were used to ensure participants understood the instructions, and participants could not proceed until they answered correctly (see S1 Appendix for details). After the game, the participants were asked to complete post-game check questions and assess the effectiveness of our manipulation of time pressure and harm (see S1 Appendix for details). The experimental flow diagram is shown in Fig 1.

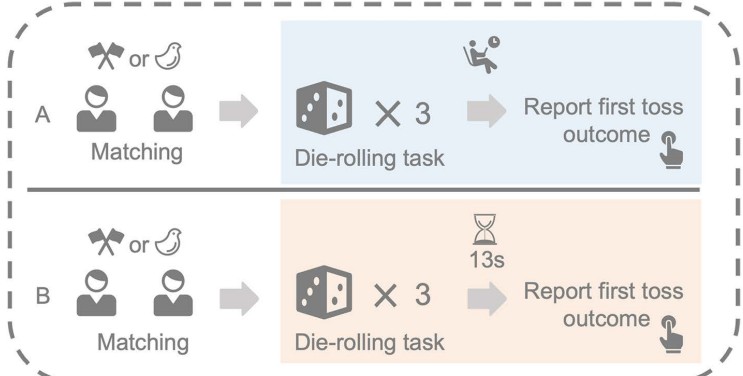

**Fig 1. The flow diagram of the experiment.** In the matching phase, participants will be randomly assigned to one of four conditions (time pressure with concrete victim, time pressure with abstract victim, self-paced with concrete victim, and self-paced with abstract victim). A) the participants could complete the task and report the first dice point with no time limit. B) participants need to complete the task and report the number of the first die within 13 seconds.

## 2.3. Measures

*Moral Identity Measure (MIM).* The MIM [29] consists of two different scales labeled internalization and symbolization. Whereas internalization aims to capture the self-importance of moral identity as a personal striving (e.g., I strongly desire to have these characteristics), symbolization focuses on overtly demonstrating these characteristics to others (e.g., I am actively involved in activities that communicate to others that I have these characteristics). Internalization is the more commonly used scale and was found to generate more consistent research findings [30], that our study only used the internalization scale. It consists of 10 statements using a 5-point response format (1 = strongly disagree, 5 = strongly agree). The scale demonstrated good reliability (alpha = 0.77) in the current sample.

*Machiavellianism (Mach-IV).* The MachIV [31] is a 20-item self-report measure of Machiavellian personality traits. Participants provided ratings on a Likert scale from 1 (strongly disagree) to 5 (strongly agree) for statements about various opinions and strategies for dealing with other people, such as "The best way to handle people is to tell them what they want to hear" and "It is wise to flatter important people". Ten items were reverse scored, such that higher scores represent higher Machiavellianism, with total scores used in the analysis. The Cronbach's alpha coefficient of Mach-IV was 0.84 in the current sample.

*The Sensitivity to Punishment and Sensitivity to Reward Questionnaire (SPSRQ).* The SPSRQ [32] is a 48-item dichotomous Yes/No self-report measure purported to load on two factors, Sensitivity to Punishment (SP) and Sensitivity to Reward (SR). The behavioral activation system and behavioral inhibition system should be measured independently. The version we used was introduced by Guo et al. [33] and revised according to the Chinese background, forming a Chinese version of the SPSRQ, qualified reliability (0.66–0.76) and validity. It contains two independent dimensions, punishment sensitivity (SP) and reward sensitivity (SR). The punishment sensitivity dimension includes 19 items, and the reward sensitivity dimension includes 12 items, all of which adopt the two-point scoring method (Yes/No). 12-item reward sensitivity was used in this study, and the scale has good reliability (alpha = 0.71) in this study.

## 2.4. Data analysis

Data were coded and analyzed using IBM SPSS Statistics 29 (IBM Corp., Armonk, NY, USA) with a significance set at $p < 0.05$. Below, we distinguish preregistered from non-preregistered exploratory analyses.

**2.4.1. Preregistered analysis.** According to our preregistration, the primary analysis was a binary logistic regression to examine the main research question regarding how time pressure and harm type influence cheating behavior. Cheating behavior (0 = cheat, 1 = honesty) served as the dependent variable, with time pressure, harm type, and their interaction as predictors.

In addition, the *t*-tests assessed the preregistered manipulations' effects on participants' subjective experience of time pressure and whether participants understood that their results in the game would (or would not) affect another participant's earnings.

**2.4.2. Non-preregistered analyses.** Beyond the preregistered plan, we conducted additional exploratory analyses to further investigate potential mechanisms. Specifically, hierarchical multiple logistic and linear regression analyses were employed to examine the influence of individual traits (moral identity, Machiavellianism, and reward sensitivity, which we mentioned in the preregistered report) on cheating behavior and on the magnitude of cheating, respectively. For these analyses, time pressure and harm type were included as binary predictors, individual trait measures as continuous predictors, and the first toss point was entered as a covariate, given its role in constraining opportunities for dishonesty. Interaction terms were entered in a subsequent step to test for moderation effects.

In addition to the classical statistical inference, and not preregistered, we also used Jeffreys-Zellner-Siow (JZS) Bayes factors (BFs) (scale R = 0.707, Rouder et al., 2009) as an alternative and/or supplementary statistical method. BF is an important method for model comparison and hypothesis testing in Bayesian statistics. BFs are applied to quantify and compare the support evidence for both null and alternative hypotheses for each contrast [34,35]. BF value reflects the

likelihood ratio between the alternative and the null. In this study, we reported $BF_{10}$ for favoring the alternative hypothesis or $BF_{01}$ for favoring the null hypothesis. The BFs in this study is calculated by the open software JASP (Version 0.18.3, https://jasp-stats.org/, JASP team, 2024). This is particularly important in our study, where null or counterintuitive findings are central. While *p*-values only indicate a failure to reject the null hypothesis, Bayes factors complement this by quantifying whether the evidence actually supports the null hypothesis, or whether the results simply reflect insufficient evidence for the alternative, thereby providing a more nuanced interpretation of our results. Although not preregistered, we believe that including Bayes factors enhances the transparency and informativeness of our findings.

## 3. Results

### 3.1. Preregistered analyses

**3.1.1. Manipulation check.** Participants in the time-pressure condition took less time to complete the task ($M = 10.44$ s, $SD = 1.76$ s) than those in the self-paced condition ($M = 14.09$ s, $SD = 4.85$ s), $t(762) = -13.82$, $p < .001$, Cohen's $d = 1.00$, 95% CI = [0.85, 1.15], $BF_{10} = 3.10 \times 10$ [36], Fig 2A shows the distribution details of the total time in each stage of the task. Furthermore, self-report results showed that participants feel more time pressure ($M = 2.65$, $SD = 1.18$) in the time pressure condition than participants in the self-paced group ($M = 2.39$, $SD = 1.09$), $t(757.51) = 3.19$, $p = 0.001$, Cohen's $d = 0.23$, 95%C.I. = [0.37, 0.09], $BF_{10} = 11.73$. These results indicated that the time-pressure manipulation was successful. Fig 2A shows the participants' post-check question scores distribution for the time-pressure and self-paced conditions.

In addition, participants in the concrete victim group ($M = 2.92$, $SD = 1.29$) believed that their performances harmed the benefits of other participants more than participants in the abstract victim group did ($M = 2.31$, $SD = 1.28$), $t(761.99) = 6.54$, $p < 0.001$, Cohen's $d = 0.47$, 95%C.I. = [0.62, 0.33], $BF_{10} = 6.34$, indicating that the harm manipulation was also successful. The participants' post-check question scores distribution in the concrete harm and abstract-harm conditions are shown in Fig 2A.

**3.1.2. Effect of time pressure and harm type on cheating.** As was shown in Fig 2B, the within-condition cheating rates were 30.89% in the time pressure with concrete victim group, 23.04% in time pressure with abstract victim group, 29.32% in self-pace with concrete victim group, and 20.42% in the self-pace with abstract victim group. Fig 2C presents the distribution histogram of time used by the time pressure group and the self-paced group across different stages of the task. Our central interest is to examine how time pressure and harm type influence participants' *cheating behavior*, a logistic regression analysis was conducted with *cheating behavior* as the predicted variable (0 = cheat, 1 = honesty), time pressure, harm type and the interaction between time pressure and harm type as the predictors. The full model was not significant, $\chi^2(760) = 7.50$, Nagelkerke $R^2 = .014$, $p = .058$, $BF_{01} = 4.32$. Further inspections showed that participants in the time-pressure condition ($M = 0.27$, $SD = 0.44$) did not lead to more cheating than participants in the self-paced condition ($M = 0.25$, $SD = 0.43$), $B = 0.15$, $SE = 0.25$, Wald (1) = 0.38, $p = .535$, OR = 1.17, 95% C.I. = [0.72, 1.90]. Interestingly, we found participants cheated more in the concrete harm condition ($M = 0.30$, $SD = 0.46$) than abstract harm condition ($M = 0.22$, $SD = 0.41$), $B = 0.48$, $SE = 0.24$, Wald (1) = 4.02, $p = .045$, OR = 1.62, 95% C.I. = [1.01, 2.59]. Importantly, inconsistent with our prediction, the interaction between time pressure and harm types was also not significant, $B = -0.08$, $SE = 0.33$, Wald (1) = 0.06, $p = .812$, OR = 0.92, 95% C.I. = [0.48, 1.78].

### 3.2. Non-preregistered analyses

**3.2.1. The influence of individual traits on time pressure and harm type in cheating behavior.** To explore the influence of individual traits on time pressure and harm type in cheating behavior. We conducted a hierarchical logistic regression with *cheating behavior* as the predicted variable (0 = cheat, 1 = honesty), the first toss point was entered on the first step (The initial die roll was critical in shaping participants' opportunities for dishonest reporting: a roll of 1 permitted over-reporting of up to 5 points, whereas a roll of 5 restricted dishonesty to just 1 point. Consequently, we included the first

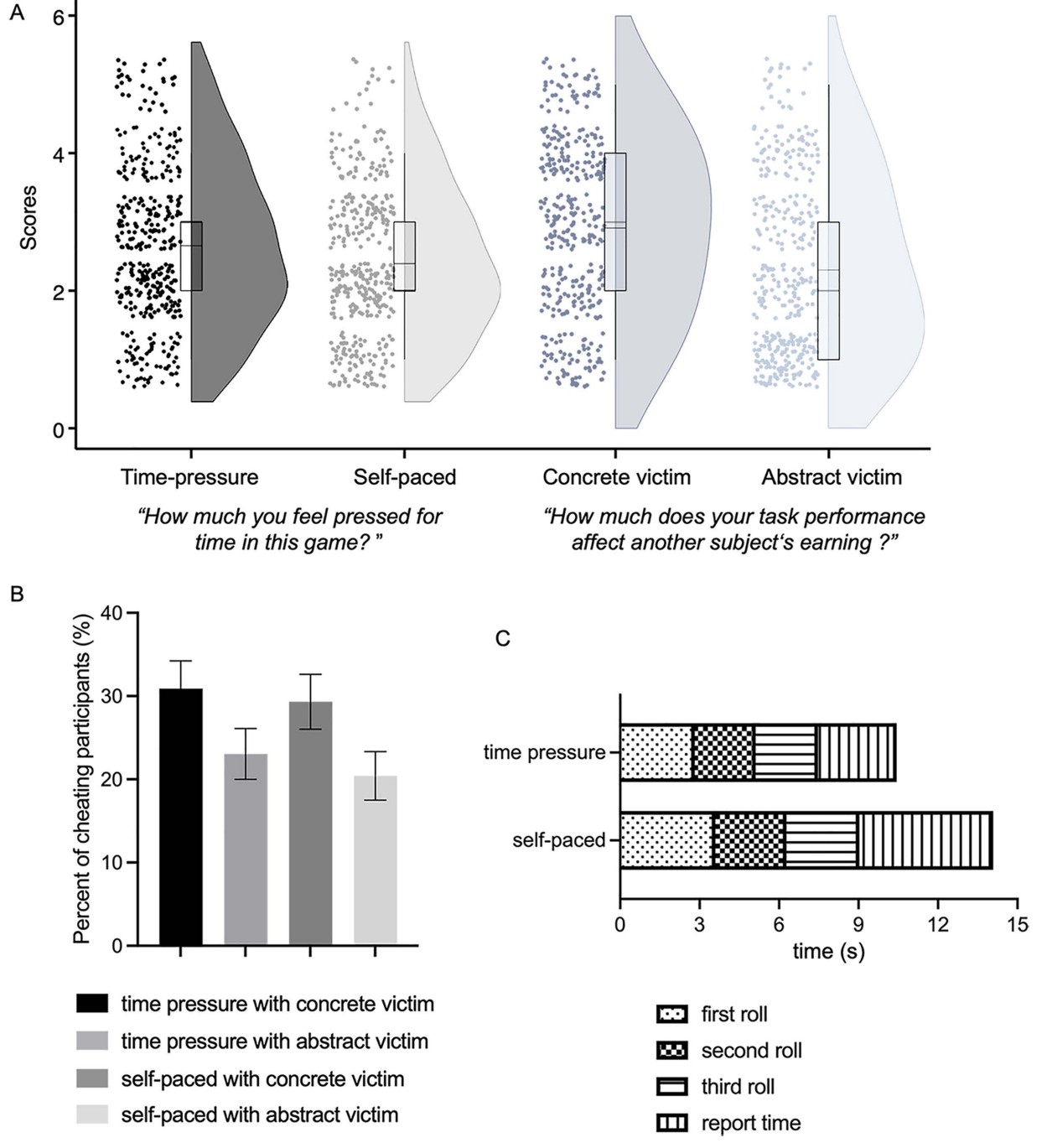

**Fig 2. A) Violin plot of rating scores (1=not affected at all; to 5=extremely affected) on manipulation check questions for time-pressure and self-paced conditions, concrete harm and abstract harm conditions.** Rectangular boxes represent the interquartile range of the distribution; the horizontal line in the middle represents the mean. The width of each plot shows the density of the data. B) The bar chart of the cheating rates in the four conditions. C) Distribution histogram of time used by time pressure group and self-paced group across different stages of the task.

die value as a covariate in our analyses), the time pressure, harm type and three scales (*MIM, Mach-IV, SPSRQ*) were entered on the second step, and the interactions were entered on the third step for predictors.

The results revealed that model 1 was significant, $\chi^2(1)$ = 54.73, Nagelkerke $R^2$ = 0.11, $p < .001$, suggesting that the first toss point negatively predicted *cheating behaviors* in the toss task, $B = -0.47$, $SE = 0.07$, Wald (1) = 46.05, $p < .001$, OR = 0.63, 95% C.I. = [0.55, 0.72].

As Fig 3A shows that the higher the point of the first dice rolled by the participant, the less likely they are to cheat, while the smaller the point rolled by the participants for the first time, the more lies the participants have. And the model 2 was also significant, $\chi^2(1)$ = 93.49, Nagelkerke $R^2$ = .17, $p < .001$. We found that harm type can positively predict cheating behavior, $B = 0.22$, $SE = 0.09$, Wald (1) = 6.21, $p = .013$, OR = 1.25, 95% C.I. = [1.05, 1.48], participants are more likely to cheat when there is a concrete harm object than when there is abstract harm. Meanwhile, the reward sensitivity ($B = 0.25$, $SE = 0.10$, Wald (1) = 5.98, $p = .014$, OR = 1.28, 95% C.I. = [1.05, 1.56]) and the March-IV ($B = 0.43$, $SE = 0.13$, Wald (1) = 11.06, $p = .001$, OR = 1.54, 95% C.I. = [1.19, 1.98]) can positively predict cheating behaviors. The model 3 was also significant, $\chi^2(1)$ = 114.09, Nagelkerke $R^2$ = .20, $p < .001$. We only found that the interaction of Moral identity with Time pressure can positively predict cheating behavior, $B = 0.27$, $SE = 0.14$, Wald (1) = 3.98, $p = .046$, OR = 1.31, 95% C.I. = [1.01, 1.71]. Hence, we categorized the data into high and low moral identity groups based on scores within one standard deviation of the mean and conducted separate logistic regression analyses for each group to explore the impact of moral identity and time pressure on cheating behavior. The results showed that the people with high moral identity are more likely to cheat under time pressure condition ($B = 1.22$, $SE = 0.42$, Wald (1) = 8.40, $p = .004$, OR = 3.38, 95%C.I. = [1.48, 7.69]). But no difference was found in the low moral identity between the time pressure and self-paced condition ($B = -0.26$, $SE = 0.46$, Wald (1) = 0.31, $p = .581$, OR = 0.78, 95%C.I. = [0.31, 1.92]). From Fig 3B, we can find that the cheating rate among participants in the time pressure group tends to increase as the level of morality increases, while the opposite trend is observed in the self-paced group. There were no other significant results (all $ps > 0.1$), see Table 1 for details.

**3.2.2. The influence of individual traits on time pressure and harm type in cheating magnitude.** This model aims to explore whether time pressure and harm type have an impact on the cheating magnitude. *Cheating magnitude* here represents how large dishonesty does those liars tend to make, was computed using the points participants reported minus the points the real toss. As is shown in Fig 3C, time pressure with concrete victim ($M \pm SE = 1.14 \pm 0.13$), the time pressure with abstract victim ($M \pm SE = 0.79 \pm 0.12$), self-pace with concrete victim ($M \pm SE = 1.00 \pm 0.12$), and self-pace with abstract victim ($M \pm SE = 0.70 \pm 0.11$). To further explore how time pressure and harm type influence *cheating magnitude*. An exploratory hierarchical linear regression analysis was conducted with *cheating magnitude* as the dependent variable. The first toss point was entered on the first step, the time pressure, harm type and three scales (*MIM, Mach-IV, SPSRQ*) were entered on the second step, the interactions were entered on the third step.

The results showed that the first model was significant, $\Delta F$ (1, 762) = 95.00, $\Delta R^2$ = 0.11, $p < 0.001$ (block 1), suggesting that the first toss point can negatively predict *cheating magnitude* in the toss task ($Bate = -0.34$, $p < .001$). As shown in Fig 3A, people tended to tell a bigger lie when the point of the dice they rolled was smaller. The second model was significant, $\Delta F$ (5, 757) = 6.38, $\Delta R^2$ = 0.14, $p < 0.001$, further inspection showed that that harm type can positively predict the *cheating magnitude* ($Bate = 0.08$, $p = .013$), participants are more likely to tell a bigger lie when there is a concrete harm object than when there is an abstract harm object. Meanwhile, the March-IV also played a positive predictive role in the toss task ($Bate = 0.15$, $p = .002$), which suggests that highly Machiavellians are more likely to tell a big lie. In addition, the third model was significant, $F$ (10, 747) = 2.14, $\Delta R^2$ = 0.17, $p = 0.019$, but further inspection revealed that none of the interactions significantly predicted *cheating magnitude,* see Table 2 for details.

## 4. Discussion

Are we intuitively inclined to cheat, or rather to be honest? A series of studies put this hypothesis to the empirical test, but found inconsistent results [5,6,10]. The social harm theory was proposed as a possible explanation, putting forward social

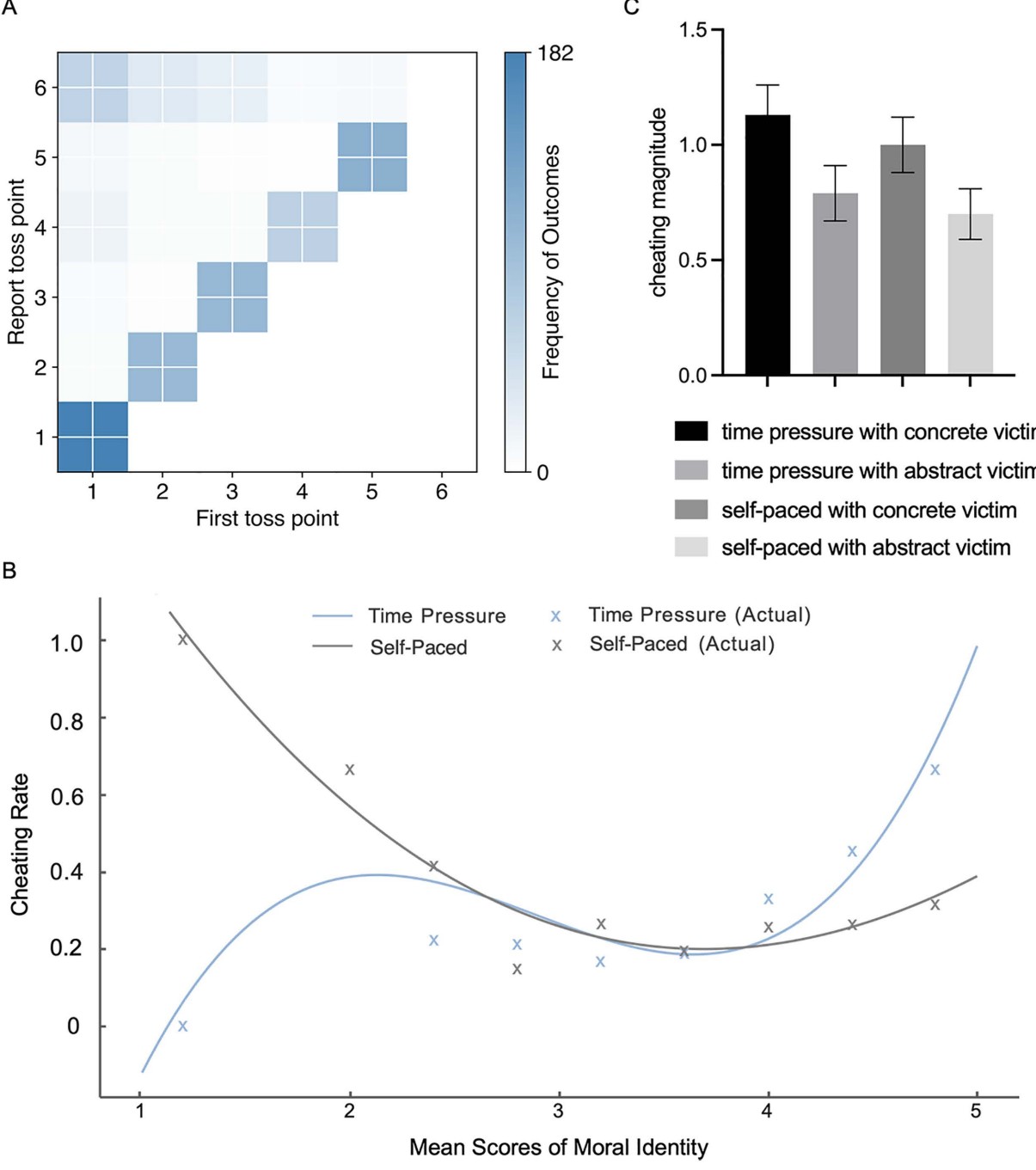

**Fig 3. A) The heatmap of the first toss point and the report toss point.** B) The ratio of cheating to honesty across the participants over morality identity under time pressure and self-paced conditions. C) The bar chart of the cheating magnitude in the four conditions.

harm as a moderator to explain the divergent findings [18], but empirical evidence supporting this theory is lacking. This pre-registration study aimed to provide well-powered evidence for this theory directly tested this theory by using the time pressure method with a total of 764 valid participants. Results showed no significant interaction between time pressure

**Table 1. Hierarchical logistic regression model for predictors of cheating behavior.**

| | Predictor | B | SE | Wald | p | Odds Ratio [95% C.I.] | $\chi^2$ | p |
|---|---|---|---|---|---|---|---|---|
| Step 1 | | | | | | | 54.73 | <.001 |
| | First toss point | −0.47 | 0.07 | 46.05 | <.001 | 0.63 [0.55, 0.72] | | |
| Step 2 | | | | | | | 93.49 | <.001 |
| | Time pressure | 0.08 | 0.09 | 0.84 | 0.359 | 1.09 [0.91, 1.29] | | |
| | Harm type | 0.22 | 0.09 | 6.21 | 0.013 | 1.25 [1.05, 1.48] | | |
| | Moral identity | −0.14 | 0.13 | 1.26 | 0.262 | 0.87 [0.67, 1.11] | | |
| | Reward sensitivity | 0.25 | 0.10 | 5.98 | 0.014 | 1.28 [1.05, 1.56] | | |
| | Machiavellianism | 0.43 | 0.13 | 11.06 | 0.001 | 1.54 [1.19, 1.98] | | |
| Step 3 | | | | | | | 114.09 | <.001 |
| | Time pressure × Harm type | −0.07 | 0.09 | 0.58 | 0.448 | 0.93 [0.78, 1.12] | | |
| | Moral identity × Time pressure | 0.27 | 0.14 | 3.98 | 0.046 | 1.31 [1.01, 1.71] | | |
| | Moral identity × Harm type | 0.03 | 0.14 | 0.04 | 0.844 | 1.03 [0.79, 1.34] | | |
| | Moral identity × Time pressure × Harm type | −0.16 | 0.14 | 1.40 | 0.237 | 0.85 [0.65, 1.11] | | |
| | Reward sensitivity × Time pressure | −0.14 | 0.11 | 1.71 | 0.191 | 0.87 [0.71, 1.07] | | |
| | Reward sensitivity × Harm type | −0.05 | 0.11 | 0.20 | 0.657 | 0.96 [0.78, 1.17] | | |
| | Reward sensitivity × Time pressure × Harm type | −0.04 | 0.11 | 0.16 | 0.693 | 0.96 [0.78, 1.18] | | |
| | Machiavellianism × Time pressure | 0.15 | 0.13 | 1.26 | 0.261 | 1.16 [0.89, 1.51] | | |
| | Machiavellianism × Harm type | −0.04 | 0.13 | 0.07 | 0.794 | 0.97 [0.74, 1.26] | | |
| | Machiavellianism × Time pressure × Harm type | 0.16 | 0.13 | 1.42 | 0.233 | 1.17 [0.90, 1.53] | | |

*Note.* Step 1: Nagelkerke $R^2$ = 0.10, Hosmer and Lemeshow Test $\chi^2(3)$ = 2.79, p = .426; Step 2: Nagelkerke $R^2$ = 0.17, Hosmer and Lemeshow Test $\chi^2(8)$ = 13.73, p = .089; Step 3: Nagelkerke $R^2$ = 0.20; Hosmer and Lemeshow Test $\chi^2(8)$ = 1.56, p = .992.

**Table 2. Hierarchical linear regression model for predictors of cheating magnitude.**

| | Predictor | Bate | t | p | ΔF | ΔR² | df | p |
|---|---|---|---|---|---|---|---|---|
| Block1 | | | | | 95.00 | 0.11 | (1, 762) | <.001 |
| | First toss point | −0.33 | −9.75 | <.001 | | | | |
| Block 2 | | | | | 6.38 | 0.14 | (5, 757) | <.001 |
| | Time pressure | 0.05 | 1.40 | 0.163 | | | | |
| | Harm type | 0.08 | 2.47 | 0.014 | | | | |
| | Moral identity | −0.07 | −1.41 | 0.160 | | | | |
| | Machiavellianism | 0.06 | 1.70 | 0.089 | | | | |
| | Reward sensitivity | 0.17 | 3.33 | 0.001 | | | | |
| Block 3 | | | | | 2.14 | 0.15 | (10, 747) | =.019 |
| | Time pressure × Harm type | −0.02 | −0.50 | 0.614 | | | | |
| | MIM × Time pressure | 0.09 | 1.73 | 0.084 | | | | |
| | MIM × Harm type | −0.02 | −0.43 | 0.669 | | | | |
| | MIM × Time pressure × Harm type | −0.07 | −1.36 | 0.173 | | | | |
| | Mach IV × Time pressure | −0.04 | −1.16 | 0.247 | | | | |
| | Mach IV × Harm type | −0.01 | −0.29 | 0.773 | | | | |
| | Mach IV × Time pressure × Harm type | −0.01 | −0.35 | 0.726 | | | | |
| | SPSRQ × Time pressure | 0.08 | 1.59 | 0.113 | | | | |
| | SPSRQ × Harm type | 0 | −0.003 | 0.997 | | | | |
| | SPSRQ × Time pressure × Harm type | 0.09 | 1.76 | 0.079 | | | | |

and harm type on cheating behavior, with the Bayesian analyses showing that the data are about 4 times more likely under the null hypothesis than under the model with the interaction. This finding does not support the proposed social harm theory. Two potential explanations can be considered to account for these results: (a) limitations in the methodology employed in the current study, (b) the volatile empirical basis for the theoretical framework.

The first possibility is that methodological changes in the current study produced these results. Our study adopts an online experiment as a means of data collection, which may have diminished the *interpersonal impact* on cheating behavior. Previous studies suggest that the salience of interpersonal impact significantly influences the manifestation of dominant impulses in socially desirable or undesirable behavior [18,37–39]. Specifically, individuals exhibit a greater reluctance to inflict the identified victims compared to unidentified victims, primarily due to the heightened potential for inducing profound emotional distress by the former group [40,41]. In the present study, participants were matched with an online "real" player to engage in the game and the concrete harm group was informed that their monetary gains would inversely affect the other party's earnings. In Pitesa's study, for instance, participants were allowed to directly obtain rewards from a jar. They were informed that all participants would receive money from the jar, implying that taking too much could reduce resources for others. Results from the manipulation check in our study indicated that participants were aware that their behavior would harm the other player, but that the harm effect is not as strong as the effect online by Pitesa et al. [19] (effect size ($\eta_p^2$) is 0.403). Therefore, the weakening of the salience of interpersonal impact in our study may be one methodological reason why we did not find the interaction between time pressure and the harm type.

A further consideration is that the issue of lying without being detected is particularly salient in online experiments. In classic offline paradigms, participants could observe the die outcome privately and thus plausibly misreport without fear of being caught [10]. By contrast, in our online setting, participants may have suspected that we recorded the actual roll, thereby enhancing the sense of exposure that cheating could be detected. Moreover, a prior multi-lab replication work has shown that participants in online often display lower engagement and a diminished sense of "experimental realism," which can weaken the effectiveness of manipulations and attenuate observed effects [42]. Online experiments inevitably raise concerns about being monitored/detected and feeling less engagement in the task. While no solution is perfect, several reasonable approaches have been proposed and adopted, including the methods used in our study. In recent years, online experiments have increasingly incorporated methodological safeguards such as attention checks [43,44] and comprehension questions [45]. Empirical evidence further supports that, when such measures are in place, online studies can achieve high levels of validity and reliability [46,47]. However, in offline settings, participants may fear that their decisions are not anonymous, such that lying could threaten their public image of honesty toward the experimenters and thereby increase the psychological cost of dishonesty [21]. By contrast, the heightened anonymity of online settings may reduce such concerns, making it easier for participants to cheat without inhibition.

The second possible explanation is the volatile empirical basis of the theory. For instance, previous studies in the Koblis meta-analyses often used behavior priming or ego depletion [48,49], but whether such manipulations can influence a significant impact on behavior is now highly contested with effect sizes often close to zero (e.g., [22]). That makes us cast doubt on their ability to influence cheating behavior. The exception is time pressure, which as a manipulation of inducing more automatic vs more strategic behavior seems uncontroversial [11,18,50]. Yet the impact of time pressure on cheating behavior shows inconsistent results [9,10]. Given the uncertainty surrounding these manipulations, it is imperative to adopt preregistered procedures to directly test and replicate their effects. This approach is paramount for strengthening the integrity and reliability of research findings in this area [51,52].

Another possibility, as suggested, is that participants in the concrete victim condition perceived the task more as a competitive or game-like context, in which "trying to win" became normatively acceptable. In contrast, the abstract victim condition may not have evoked the same framing, leading to relatively less cheating. Relatedly, Moore et al. [53] highlighted that many everyday moral dilemmas can be understood as conflicts between benevolence (compassion for a specific individual [54]) and integrity (adherence to impartial rules [54,55]). Within this framework, cheating in the abstract

victim condition may be interpreted as a failure of integrity (violating a general rule), whereas cheating in the concrete victim condition may reflect a lack of benevolence (disregarding the welfare of an identifiable other). This perspective helps clarify why the same dishonest behavior can be evaluated differently across conditions.

We observed that certain personality traits have a potential influence on individuals' propensity for engaging in dishonest behavior. In our exploratory analyses, we found an interaction between moral identity and time pressure. Participants in the time pressure group tend to increase as the level of morality increases, while the opposite trend is observed in the self-paced group. This finding contradicts previous research by Xu and Ma [24], who found that moral identity moderated the effect of ego depletion on cheating behavior. One possible explanation is that time pressure alters how individuals with high moral identity process the situation. Although moral identity is generally expected to show a "moral default" [56,57], and reduce dishonest behavior, under time pressure participants may rely more on intuitive or heuristic responses rather than deliberate moral reasoning. In our task, the 13-second limit may have prompted even those with high moral identity to default to such heuristics, resulting in more cheating. Thus, moral identity may buffer dishonesty only when sufficient time is available for reflection. These findings should be interpreted with caution, and further research is needed to clarify the mechanisms underlying this counterintuitive interaction. Furthermore, Campos and colleagues demonstrated that time pressure affects dishonest behavior differently depending on the level of time pressure evaluated [58]. Additionally, consistent with previous research, individuals with high reward sensitivity exhibited challenges in resisting temptations, whereas those with low reward sensitivity demonstrated easier in resisting high reward incentives. Furthermore, the study found a positive correlation between Machiavellianism and cheating behavior, although this association did not reach statistical significance when considering the magnitude of cheating. It is possible that the limited payout offered in this study (up to 6 RMB) may have constrained the ability to observe a distinct difference in the magnitude of cheating behavior.

Finally, it is important to recognize that dishonesty can take multiple forms, including self-serving, prosocial, and altruistic lies. The present research only focuses on lying for self-profits which is based on previous research. Future studies should examine whether prosocial lies are intuitive or deliberative. At the same time, we acknowledge that the broader literature emphasizes the diversity of lying motives and their links to (pro) social behavior [59,60]. Future research would benefit from developing improved paradigms that examine prosocial and altruistic lies more directly, with a particular focus on the motivations underlying dishonesty and on how its consequences—whether positive, negative, or neutral—shape behavior.

In sum, our findings question the social harm theory and its empirical basis. To understand whether and when people are intuitively (dis-)honest, we require more robust empirical research.

## Supporting information

**S1 Appendix. The pre- and post-game questions.**
(DOCX)

## Author contributions

**Conceptualization:** Jiayu Cheng, Chongxiang Wang, Liyang Sai.

**Data curation:** Jiayu Cheng, Yue Liu.

**Formal analysis:** Jiayu Cheng.

**Funding acquisition:** Liyang Sai.

**Investigation:** Jiayu Cheng, Haoran Wang, Yue Liu, Chongxiang Wang.

**Methodology:** Jiayu Cheng, Haoran Wang, Chongxiang Wang.

**Project administration:** Jiayu Cheng.

**Software:** Jiayu Cheng, Haoran Wang.

**Supervision:** Jiayu Cheng, Qingzhou Sun, Bruno Verschuere, Liyang Sai.

**Validation:** Jiayu Cheng, Haoran Wang, Yue Liu.

**Visualization:** Jiayu Cheng.

**Writing – original draft:** Jiayu Cheng, Haoran Wang, Yue Liu.

**Writing – review & editing:** Jiayu Cheng, Haoran Wang, Yue Liu, Chongxiang Wang, Qingzhou Sun, Bruno Verschuere, Liyang Sai.

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
