## [Decision Letter · Decision Letter 0]

12 Aug 2025

I have now received the reports from two reviewers with considerable knowledge and expertise in your topic domain (their detailed feedback appears below or as separate files). On the positive side, both the reviewers find your manuscript easy to follow and believe that you have interesting results based on a rigorous and preregistered empirical investigation. On the more critical side, however, they also raise a series of substantial concerns, which largely focus on lacking recruitment details, potential differential attrition across conditions, insufficient level of detail in the preregistered analyses, and clarity issues with respect to what was and was not preregistered. Moreover, they also note construct validity/confusion concerns, a failure to clearly discuss certain seemingly counterintuitive findings (e.g., more cheating in the concrete victim condition), and a lack of elaboration pertaining to whether, why, and how you believe that the study setting (online vs. real life) might have influenced your results, potentially due to lacking engagement among online participants (Baumeister et al., 2023). Further, the reviewers mention some uncited work which might bolster your theorizing and storyline.

Based on the constructive comments from the reviewers and my own reading of your paper, I am willing to move this manuscript into a second round of reviews. Given the magnitude of some of the issues identified by the reviewers, this will be a major revision, although a revision with a relatively clear path toward publication as long as you meticulously address all the substantive concerns flagged in the separate reviewer reports. Please make sure to reply to all comments made by the reviewers, incorporate needed changes in the manuscript, and then send an updated version of it along with your revision notes at your earliest convenience. Try to do this within the next two months. If you need additional time, feel free to let me know (at tobias.otterbring@uia.no) and I am happy to extend your revision window.

Reference

Baumeister, R. F., Tice, D. M., & Bushman, B. J. (2023). A review of multisite replication projects in social psychology: is it viable to sustain any confidence in social psychology’s knowledge base? Perspectives on Psychological Science, 18(4), 912-935.

    • A marked-up copy of your manuscript that highlights changes made to the original version. You should upload this as a separate file labeled 'Revised Manuscript with Track Changes

We look forward to receiving your revised manuscript.

Kind regards,

Tobias Otterbring

Handling Editor, PLOS One

Journal Requirements:

[This work was supported by the National Natural Science Foundation of China (32271111, U1736125 to L. Sai); the Science and Technology Innovation 2030-“Brain Science and Brain-like Research” Major Project (Grant/Award Number: 2022ZD0210800).].

[This work was supported by the National Natural Science Foundation of China (32271111, U1736125 to L. Sai); the Science and Technology Innovation 2030-“Brain Science and Brain-like Research” Major Project (Grant/Award Number: 2022ZD0210800).]

[This work was supported by the National Natural Science Foundation of China (32271111, U1736125 to L. Sai); the Science and Technology Innovation 2030-“Brain Science and Brain-like Research” Major Project (Grant/Award Number: 2022ZD0210800).]

Reviewers' comments:

Reviewer's Responses to Questions

**Comments to the Author**

1. Is the manuscript technically sound, and do the data support the conclusions?

Reviewer #1: Yes

Reviewer #2: Yes

2. Has the statistical analysis been performed appropriately and rigorously?

Reviewer #1: Yes

Reviewer #2: Yes

3. Have the authors made all data underlying the findings in their manuscript fully available?

Reviewer #1: Yes

Reviewer #2: Yes

4. Is the manuscript presented in an intelligible fashion and written in standard English?

Reviewer #1: Yes

Reviewer #2: Yes

Reviewer #1: Hello

I have read your manuscript "Intuitive or Deliberative Dishonesty: The effect of abstract versus concrete victim" submitted to Plos One. Below is my review.

First, I applaud the fact that you conducted a preregistered well-powered study that aims to replicate earlier findings. This is surely a merit.

Second, for the most part I think the paper is well written and easy to follow. You cite plenty of relevant research. Perhaps you want to look on Josh Greenes work on dual process morality where it is proposed that intuition leads to deontological decision making (following rules such as not harming) whereas deliberation leads to utilitarian decision making (maximizing good consequences). See Greene, J. D. (2008). The secret joke of Kant's soul. In W. Sinnott-Armstrong (Ed.), Moral psychology, Vol 3: The neuroscience of morality: Emotion, brain disorders, and development. (pp. 35-80). Cambridge: MIT Press. for an overview.

I wonder whether you measure dishonestly or selfishness. In your study, lying is always selfish as it increases the participants’ payoff so dishonestly and selfishness are difficult to disentangle. Still, it is important to remember that both lies and truth telling can have both positive and negative (or no) consequences for both the person deciding whether to lie, and for the person being lied to. There are white lies, prosocial lies and altruistic lies. This should be highlighted and you should discuss the related literature investigating about whether prosociality is intuitive or deliberate. For instance Rand, D. G., Greene, J. D., & Nowak, M. A. (2012). Spontaneous giving and calculated greed [10.1038/nature11467]. Nature, 489(7416), 427-430. Tinghög, G., Andersson, D., Bonn, C., Böttiger, H., Josephson, C., Lundgren, G., Västfjäll, D., Kirchler, M., & Johannesson, M. (2013). Intuition and cooperation reconsidered. Nature, 498(7452), E1-E2.

I am not perfectly updated on the literature, but I feel that the fact that you did the die roll and reporting online rather than in real life can influence the believability of being able to lie without being detected. In the original die rolling studies, participants rolled a die in private and then reported the results on a separate paper (correct me if I am misremembering). This approach made it believable that one could cheat without being detected. In your online study, I feel that I would assume that you recorded the result of the die roll and could easily “prove that I lied” if you wanted to. You might not agree, but I feel you should discuss whether this rather big methodological difference could explain diverging results.

Relatedly, could it even be that what you measure is not dishonestly (lying in the belief that no one notices) but rather how much participants care about being caught lying?

I like your method, but did you consider manipulating the die roll results rather than just observing them. In an online paradigm, this could easily be done I suppose and it would give you more control. Not saying that it must be done, but that it could be reflected upon.

I believe the most surprising result is that people cheated more in the concrete victim conditions. This seems to be an effect in the opposite direction of what could be expected and against the identifiable victim effect. I strongly suggest that you discuss and try to make sense of this result. Not only a null effect but an effect in the opposite direction. Could it be that participants in the concrete victim conditions perceived it more like a game where the normative response was to “try to win” whereas participants in the abstract condition did not. Not sure, but in your version it seems like you want to hide this significant effect.

I do not really understand panel C on Figure 2. The x-axis display the two manipulated variables, but what does “score” on the Y-axis stand for?

The weak interaction effect of moral identity and time pressure condition is interesting but hard to make sense of. It makes sense that low moral ID and no time pressure leads to much cheating, but it does not makes much sense that time pressure leads to much cheating among those with high moral id. Or does it? I feel that you report results in a good way, but you could do more to at least offer possible reasons for the obtained results.

The study below seems relevant. I feel that cheating in the abstract victim condition is to act with low integrity (break a rule, and to act unfairly) wheres cheating in the concrete victim condition is to act with low benevolence.

Moore, A. K., Munguia Gomez, D. M., & Levine, E. E. (2019). Everyday dilemmas: New directions on the judgment and resolution of benevolence–integrity dilemmas. Social and Personality Psychology Compass, 13(7), e12472.

Reviewer #2: This paper reports a preregistered experiment investigating whether time pressure (intuition) increases dishonesty (cheating behavior) when dishonesty harms an abstract (but not concrete) other. Results do not support this prediction, as there is no significant interaction effect between time pressure (vs. no time pressure) and harm type (abstract vs. concrete victim). However, there was a main effect of harm type such that participants cheated more when the victim was concrete than abstract, opposite of expectations. The experiment also explores individual traits.

The study addresses an interesting and relevant question and I think the paper is well-organized and clear and the conclusions appear warranted by the data. I hope my comments below can help improve the paper further.

1. Some more detail about how/where participants were recruited would be desirable (e.g. was it using a recruitment platform, social media, etc.).

2. I am a bit puzzled by the results from the manipulation check for the harm type manipulation (concrete vs. abstract victim). The question asks: “How much does your task performance affect another subject’s earnings?” It seems to me that participants in the “abstract” condition should answer 1 (“not at all”) if they correctly understood the instructions, yet the mean rating is 2.31. Furthermore, one of the pre-game rule-check questions asks how much the opponent could obtain if the participant reports 5 points; given that participants in the abstract condition are not playing with an opponent, it is unclear what they should answer here. I wonder if there might be some selection effect because of this that affects the two conditions differently, as participants are screened out if they answer incorrectly. Also, how many participants in each condition failed the rule-check questions?

3. I would suggest reporting the full regression results from the main hypothesis test (section 3.1.2), considering that it is the main (and preregistered) hypothesis test.

4. In some places it could be made clearer what was and what was not preregistered among the analyses. In particular, it could be mentioned in the analysis section that the use of Bayes factors was not preregistered.

5. Figure 2c (manipulation checks) could be modified slightly to improve interpretability on its own (without having to read the text). E.g. add the manipulation check questions into the figure as subheadings or similar.

Minor:

1. Look over the grammar in this sentence: ”Other experimental works have similarly revealed … will increase self-serving dishonesty.” (p 3-4)

2. “cognitively impaired” (p 4) has a different connotation than what I think the authors intend – change to “cognitively depleted” or “temporarily cognitively impaired”?

3. Preregistrated (p 5) -> preregistered

4. Mofied -> modified (p 5)

5. Pista et al./Pistea et al. -> Pitesa et al. (p 5)

6. ”Exploratory” -> ”As exploratory analyses” or similar (p 6)

7. present study -> the present study (p 6)

8. preregistratrion study -> preregistered study (p 23)

9. P. 11: “t-tests assessed the manipulations' effects on time pressure and harm types” – I believe this refers to the manipulation checks? I suggest a slight rephrasing for clarity, e.g. “t-tests assessed the manipulations’ effects on participants’ subjective experience of time pressure and whether participants understood that their results in the game would (or would not) affect another participant’s earnings” (or whatever the authors find more appropriate).

**Do you want your identity to be public for this peer review?** For information about this choice, including consent withdrawal, please see our Privacy Policy

Reviewer #1: **Yes:** Arvid Erlandsson

Reviewer #2: No

---

## [Author Response · Author response to Decision Letter 1]

13 Oct 2025

Response Letter

Review PONE-D-25-29808

Title: Intuitive or Deliberative Dishonesty: The effect of abstract versus concrete victim

Editor comment:

Thank you for submitting your manuscript to PLOS ONE. After careful consideration, we feel that it has merit but does not fully meet PLOS ONE’s publication criteria as it currently stands. Therefore, we invite you to submit a revised version of the manuscript that addresses the points raised during the review process.

I have now received the reports from two reviewers with considerable knowledge and expertise in your topic domain (their detailed feedback appears below or as separate files). On the positive side, both the reviewers find your manuscript easy to follow and believe that you have interesting results based on a rigorous and preregistered empirical investigation. On the more critical side, however, they also raise a series of substantial concerns, which largely focus on lacking recruitment details, potential differential attrition across conditions, insufficient level of detail in the preregistered analyses, and clarity issues with respect to what was and was not preregistered. Moreover, they also note construct validity/confusion concerns, a failure to clearly discuss certain seemingly counterintuitive findings (e.g., more cheating in the concrete victim condition), and a lack of elaboration pertaining to whether, why, and how you believe that the study setting (online vs. real life) might have influenced your results, potentially due to lacking engagement among online participants (Baumeister et al., 2023). Further, the reviewers mention some uncited work which might bolster your theorizing and storyline.

Based on the constructive comments from the reviewers and my own reading of your paper, I am willing to move this manuscript into a second round of reviews. Given the magnitude of some of the issues identified by the reviewers, this will be a major revision, although a revision with a relatively clear path toward publication as long as you meticulously address all the substantive concerns flagged in the separate reviewer reports. Please make sure to reply to all comments made by the reviewers, incorporate needed changes in the manuscript, and then send an updated version of it along with your revision notes at your earliest convenience. Try to do this within the next two months. If you need additional time, feel free to let me know (at tobias.otterbring@uia.no) and I am happy to extend your revision window.

Reference

Baumeister, R. F., Tice, D. M., & Bushman, B. J. (2023). A review of multisite replication projects in social psychology: is it viable to sustain any confidence in social psychology’s knowledge base? Perspectives on Psychological Science, 18(4), 912-935.

Response: Thank you very much for your comment and for considering our manuscript for publication in PLOS ONE. We are grateful for the constructive comments provided by you and the reviewers. We truly appreciate your recognition of the strengths of our study, as well as the detailed feedback that has helped us to improve the manuscript substantially.

As will become clear in our response to the reviewers, we carefully considered all of the issues raised, including those concerning recruitment details (see page 7 in our revision draft), preregistration transparency (see pages 12 in our revision draft), potential attrition differences across conditions (see our response to Reviewer #2, Comment 2), discussion of counterintuitive findings (see pages 25-27 in our revision draft), and possible implications of conducting the study online (see pages 24-25 in our revision draft) etc. We have revised the manuscript accordingly and clarified these points in detail. We have also incorporated additional citations suggested by the reviewers to strengthen the theoretical framing.

In addition, we read Baumeister et al. (2023) and carefully considered the possible reasons for large-scale replication failures discussed by the authors, including that the original hypothesis was wrong; the hypothesis was not properly tested because of operational failure; the low engagement of participants; and bias toward failure. Although a prior multi-lab replication work has found that participants in online settings often display lower engagement and a diminished sense of “experimental realism,” which can weaken the effectiveness of manipulations and attenuate observed effects (Baumeister et al., 2023), it is also noteworthy that many online experiments have increasingly incorporated attention checks (Hauser & Schwarz, 2016; Zickfeld et al., 2025), comprehension questions (Parra et al., 2024), and other methodological safeguards, thereby gradually improving data quality (Douglas et al., 2023; Horton et al., 2011). Thus, we think that the assumption that lab participants are attentive and engaged but online participants is not that straightforward. Nevertheless, this issue underscores the importance of further refining experimental paradigms and strengthening experimental control in online settings to ensure the collection of higher-quality data. We have added a corresponding discussion of these issues in the revised manuscript (pls see pages 23-25).

In the text below, we provide a point-by-point response to all reviewer comments. We sincerely thank you and the reviewers for your constructive guidance and for giving us the opportunity to revise our work. We look forward to our revised manuscript moving to the second round of reviews.

Reviewer #1:

I have read your manuscript "Intuitive or Deliberative Dishonesty: The effect of abstract versus concrete victim" submitted to Plos One. Below is my review.

Comment 1: First, I applaud the fact that you conducted a preregistered well-powered study that aims to replicate earlier findings. This is surely a merit.

Response: Thank you for this positive comment.

Comment 2: Second, for the most part I think the paper is well written and easy to follow. You cite plenty of relevant research. Perhaps you want to look on Josh Greenes work on dual process morality where it is proposed that intuition leads to deontological decision making (following rules such as not harming) whereas deliberation leads to utilitarian decision making (maximizing good consequences). See Greene, J. D. (2008). The secret joke of Kant's soul. In W. Sinnott-Armstrong (Ed.), Moral psychology, Vol 3: The neuroscience of morality: Emotion, brain disorders, and development. (pp. 35-80). Cambridge: MIT Press. for an overview.

Response: Thank you very much for your kind feedback and the helpful suggestion regarding Greene’s work on dual process morality. We agree that his proposal—that intuitive processes tend to drive deontological decisions while deliberative processes support utilitarian reasoning—is highly influential and insightful. We did review Greene (2008) and found the perspective compelling. However, given that our study focuses specifically on (dis)honesty behavior in the context of abstract vs. concrete victims with time pressure rather than moral dilemma judgments per se, we felt that the dual-process morality framework was not the most fitting for our research aims. Still, we truly appreciate the suggestion and will consider referencing it in future related work.

I wonder whether you measure dishonestly or selfishness. In your study, lying is always selfish as it increases the participants’ payoff so dishonestly and selfishness are difficult to disentangle. Still, it is important to remember that both lies and truth telling can have both positive and negative (or no) consequences for both the person deciding whether to lie, and for the person being lied to. There are white lies, prosocial lies and altruistic lies. This should be highlighted and you should discuss the related literature investigating about whether prosociality is intuitive or deliberate. For instance Rand, D. G., Greene, J. D., & Nowak, M. A. (2012). Spontaneous giving and calculated greed [10.1038/nature11467]. Nature, 489(7416), 427-430. Tinghög, G., Andersson, D., Bonn, C., Böttiger, H., Josephson, C., Lundgren, G., Västfjäll, D., Kirchler, M., & Johannesson, M. (2013). Intuition and cooperation reconsidered. Nature, 498(7452), E1-E2.

Response: Thank you for this comment. Our research focuses on lying for self-profit. This is because the theoretical debates focus on situations where lying could bring benefits (Shalvi et al., 2012). However, we agree with the reviewer that there are other kinds of lies, such as white lies, prosocial lies and altruistic lies, and they are also worthy to examine. We have added one paragraph to discuss that future studies should also examine whether prosocial lies are intuitive or deliberative. See page 27:

“Finally, it is important to recognize that dishonesty can take multiple forms, including self-serving, prosocial, and altruistic lies. The present research only focuses on lying for self-profits which is based on previous research. Future studies should examine whether prosocial lies is intuitive or deliberative. At the same time, we acknowledge that the broader literature emphasizes the diversity of lying motives and their links to (pro) social behavior (Rand et al., 2012; Tinghög et al., 2013). Future research would benefit from developing improved paradigms that examine prosocial and altruistic lies more directly, with a particular focus on the motivations underlying dishonesty and on how its consequences—whether positive, negative, or neutral—shape behavior.”

I am not perfectly updated on the literature, but I feel that the fact that you did the die roll and reporting online rather than in real life can influence the believability of being able to lie without being detected. In the original die rolling studies, participants rolled a die in private and then reported the results on a separate paper (correct me if I am misremembering). This approach made it believable that one could cheat without being detected. In your online study, I feel that I would assume that you recorded the result of the die roll and could easily “prove that I lied” if you wanted to. You might not agree, but I feel you should discuss whether this rather big methodological difference could explain diverging results.

Response: Thank you for this valuable comment. We agree with the reviewer’s consideration that participants may think that they will be detected if they lied. In the typical laboratory version of this paradigm (see e.g., Van der Cruyssen et al., 2020), participants indeed rolled a die privately under a cup and observed the outcome through a small hole (visible only to themselves). They then reported the result, with higher reported outcomes yielding larger rewards (e.g., reporting a six earned €12). Because the outcome was known only to the participant, they had a credible opportunity to misreport without the risk of being caught. This procedure indeed allows for cheating without detection. Online implementations of this paradigm inevitably raise concerns about how anonymity is ensured. While no solution is perfect, several reasonable approaches have been proposed and adopted, including the method used in our study. Numerous related studies (e.g., Alfonso et al., 2022; Zickfeld et al., 2025) have successfully conducted this paradigm online, and meta-analyses suggest that, with appropriate controls, online results can be as reliable as those obtained in laboratory settings. In recent years, online experiments have increasingly incorporated methodological safeguards such as attention checks (Hauser & Schwarz, 2016; Zickfeld et al., 2025) and comprehension questions (Parra et al., 2024). Empirical evidence further supports that, when such measures are in place, online studies can achieve high levels of validity and reliability (Douglas et al., 2023; Horton et al., 2011). At the same time, it is important to note that online studies also offer a greater degree of anonymity and less social presence compared to laboratory settings. Previous studies suggest that the salience of interpersonal impact significantly influences the manifestation of dominant impulses in socially desirable or undesirable behavior (Gneezy, 2005; Jones, 1991; Köbis et al., 2019; Mok & De Cremer, 2018). Specifically, individuals exhibit a greater reluctance to inflict the identified victims compared to unidentified victims, primarily due to the heightened potential for inducing profound emotional distress by the former group (Kogut & Ritov, 2005; Milgram, 1965). This also highlights that online experiments offer many factors for further exploration. We call for future research to conduct comparative studies and develop new paradigms to better separate these factors. In the revised manuscript, we have expanded our Discussion about this issue (pls see page 23-25). We sincerely appreciate your suggestion, which helped us clarify this important aspect.

“The first possibility is that methodological changes in the current study produced these results…

A further consideration is that the issue of lying without being detected is particularly salient in online experiments. In classic offline paradigms, participants could observe the die outcome privately and thus plausibly misreport without fear of being caught (Van der Cruyssen et al., 2020). By contrast, in our online setting, participants may have suspected that we recorded the actual roll, thereby enhancing the sense of exposure that cheating could be detected. Moreover, a prior multi-lab replication work has shown that participants in online often display lower engagement and a diminished sense of “experimental realism,” which can weaken the effectiveness of manipulations and attenuate observed effects (Baumeister et al., 2023). Online experiments inevitably raise concerns about being monitored/detected and feeling less engaged in the task. While no solution is perfect, several reasonable approaches have been proposed and adopted, including the methods used in our study. In recent years, online experiments have increasingly incorporated methodological safeguards such as attention checks (Hauser & Schwarz, 2016; Zickfeld et al., 2025) and comprehension questions (Parra, 2024). Empirical evidence further supports that, when such measures are in place, online studies can achieve high levels of validity and reliability (Douglas et al., 2023; Horton et al., 2011). However, in offline settings, participants may fear that their decisions are not anonymous, such that lying could threaten their public image of honesty toward the experimenters and thereby increase the psychological cost of dishonesty (Gerlach et al., 2019). By contrast, the heightened anonymity of online settings may reduce such concerns, making it easier for participants to cheat without inhibition. ”

Relatedly, could it even be that what you measure is not dishonestly (lying in the belief that no one notices) but rather how much participants care about being caught lying?

Response: Thank you for raising this important point. We agree that, in principle, there is a distinction between measuring dishonesty per se and measuring how much participants care about being caught lying. Methodologically, our die-roll paradigm is designed to capture dishonesty by comparing self-reported outcomes with the actual probabilities of dice results. Participants always had an opportunity to increase their payoff by inflating their report, and since their individual rolls were private, the only way for them to gain extra points was through dishonesty. While we cannot completely rule out that some participants refrained from lying because they worried about being detected, the paradigm and incentive structure strongly support the interpretation that the primary behavior we measured was honesty versus dishonesty.

Comment 3: I like your method, but did you consider manipulating the die roll results rather than just observing them. In an online paradigm, this could easily be done I suppose and it would give you more control. Not saying that it must be done, but that it could be reflected upon.

Response: In online settings, there are indeed two ways to handle the die role outcome: manipulating it (Mazar et al., 2008) versus observing (Shalvi et al., 2011; Van der Cruyssen et al., 2020). We c

---

## [Decision Letter · Decision Letter 1]

16 Dec 2025

Intuitive or Deliberative Dishonesty: The effect of abstract versus concrete victim

PONE-D-25-29808R1

Dear Dr. Sai,

We’re pleased to inform you that your manuscript has been judged scientifically suitable for publication and will be formally accepted for publication once it meets all outstanding technical requirements.

Kind regards,

Tobias Otterbring

Academic Editor

PLOS ONE

Additional Editor Comments (optional):

Dear authors,

Thank you for delivering a responsive revision. Based on your material revisions in the manuscript and your detailed replies to the reviewers, both of whom are positive toward the current version of the manuscript, I am happy to recommend acceptance of your paper in its current form. That said, I strongly recommend you to consider the remaining minor suggestions from the reviewers in the proof process. Congratulations!

Kind regards,

Tobias Otterbring

Associate Editor, PLOS One

Reviewers' comments:

Reviewer's Responses to Questions

**Comments to the Author**

Reviewer #1: All comments have been addressed

Reviewer #2: (No Response)

2. Is the manuscript technically sound, and do the data support the conclusions?

Reviewer #1: Yes

Reviewer #2: Yes

3. Has the statistical analysis been performed appropriately and rigorously?

Reviewer #1: Yes

Reviewer #2: Yes

4. Have the authors made all data underlying the findings in their manuscript fully available?

Reviewer #1: Yes

Reviewer #2: Yes

5. Is the manuscript presented in an intelligible fashion and written in standard English?

Reviewer #1: Yes

Reviewer #2: Yes

Reviewer #1: Hello

This is reviewer 1 from the earlier round. I have read your revised manuscript "Intuitive or Deliberative Dishonesty: The effect of abstract versus concrete victim" submitted to Plos One. Below is my review.

Thank you for the good revision. I feel you addressed most of my concerns in a satisfactory way. I have some final suggestions, but after they are fixed, I am willing to recommend publication.

Page 6. I feel you repeat yourself regarding analyses about individual differences.

Page 8-9: I feel that the die-rolling task can be explained more in detail at this point (it become clearer when you go through the results). When reading about it here, it is not clear (unless one has read Shalvi et al) that the die-roll was semi-random (you controlled so that it never landed on 6 but it was random whether it landed on 1-5). Please make it clear how you operate cheating and honesty, and I would also mention cheating magnitude at this point.

Page 15 Fig 2. Here your refer to the conditions as harm vs no harm. I think it is better to refer to it as concrete vs abstract harm

Page 24. Last rows: “inflict [harm to] the identified victims”

The discussion is much improved, but I think you can separate “lack of expected results” (which can be because a lot of different factors) from “presence of unexpected results” even more explicit. Again, I am surprised that cheating was higher in the concrete harm conditions. The discussion you have is good, but I would also argue that your “concrete harm” condition could be even more concrete (e.g. you learn more individualizing aspects about the player you are paired with and perhaps even that you anticipate how you will be able to observe the reaction when that player receives his/her pay).

Reviewer #2: Thank you for responding to my comments. I think the authors have answered the points raised.

Regarding my comment about potential selection effects - thank you for clarifying that participants are not screened out based on their answers to the pre-game rule-check questions. I would suggest adjusting the wording in the paper slightly to clarify this, specifically on p. 9 where it is stated that “only those who passed the rule-check questions could continue to the formal experiment” (to something along the lines of "participants could not proceed until they answered correctly")

**Do you want your identity to be public for this peer review?** For information about this choice, including consent withdrawal, please see our Privacy Policy

Reviewer #1: **Yes:** Arvid Erlandsson

Reviewer #2: No

---

## [Editor Report · Acceptance letter]

PONE-D-25-29808R1

PLOS One

Dear Dr. Sai,

I'm pleased to inform you that your manuscript has been deemed suitable for publication in PLOS One. Congratulations! Your manuscript is now being handed over to our production team.

Kind regards,

on behalf of

Professor Tobias Otterbring

Academic Editor

PLOS One